# Health and social care workers experiences of coping while working in the frontline during the COVID-19 pandemic: One year on

Karina Soubra[1], Camilla Tamworth[2], Zeast Kamal[2], Clare Brook[3], Dawn Langdon[1], Jo Billings[2]*

1 Department of Psychology, Royal Holloway University of London, London, United Kingdom, 2 Division of Psychiatry, University College London, London, United Kingdom, 3 Acorn Group Practice, Twickenham, United Kingdom

* j.billings@ucl.ac.uk

## Abstract

### Background

The unprecedented pressure of working on the frontline during the Covid-19 pandemic had a demonstrable impact on the mental health and wellbeing of health and social care workers in the early stages of the pandemic, however, less research has focused on workers' experiences over the longer course of the pandemic.

### Aims

We set out to develop an explanatory model of the processes that helped and hindered the coping of HSCWs working over the course of the Covid-19 pandemic.

### Method

Twenty HSCWs based in the UK took part in the study. They completed semi-structured interviews 12–18 months after the peak of the first wave in the UK. Interviews were transcribed and analysed using grounded theory methodology.

### Results

The analysis identified eleven theoretical codes: *personal context*, *organisational resources*, *organisational response*, *management*, *colleagues*, *decision-making and responsibilities*, *internal impacts*, *external impactors*, *safety*, *barriers to accessing support* and *temporal factors*. The findings suggest that factors related to the individual themselves, their personal context, the organisation they work in, their managers, the support structures around them and their sense of safety impacted on HSCWs; ability to cope. Some factors changed over time throughout the first year of the pandemic, such as workload and staff illness, which further impacted HSCWs' coping. There were many barriers to accessing support that also impacted coping, including availability, awareness and time. The relationship between the factors that impacted coping are represented in an explanatory model.

**Data Availability Statement:** Data supporting the findings of this study have not been made available due to the personal and sensitive content of the frontline workers' accounts, in line with the

requirements stipulated by the UCL and Royal Holloway Research Ethics Committee. Further information can be provided by ethics@ucl.ac.uk.

**Funding:** The authors received no specific funding for this work.

**Competing interests:** The authors have declared that no competing interests exist.

## Conclusions

The findings extend previous studies on the mental health impact on frontline HSCWs working during Covid-19, providing novel insight by developing an explanatory model illustrating the underlying factors that impacted their coping experiences over the course of the pandemic in the UK. The findings from this study may assist in the development of improved and more effective support for HSCWs going forwards.

## Introduction

The Coronavirus disease 2019 (Covid-19) pandemic placed extreme demands on health and social care workers (HSCWs) around the world as they faced a novel rapidly spreading virus resulting in high numbers of patients with high mortality rates. The limited availability of effective treatment options made it difficult for healthcare systems to cope [1]. Rising cases led to HSCWs working longer hours and working in more intense environments, with the added challenges of having to follow strict infection control measures and not always having adequate personal protective equipment (PPE) [2]. Many HSCWs were redeployed and forced to work in unfamiliar settings. They also continued to work despite the risks to their own physical safety and to that of their loved ones.

As the pandemic progressed, research on the mental health and wellbeing of staff started emerging from the UK [3] and the rest of the world [4,5], which demonstrated the negative effects on mental health of working in the frontline as a HSCW during the COVID-19 pandemic Frontline HSCWs all around the world experienced elevated rates of anxiety, depression and PTSD in response to COVID-19. Potential risk factors for developing mental health disorders while working in the frontlines during COVID-19 included working directly with patients with confirmed or suspected COVID-19 and concerns about personal safety due to inadequate access to appropriate personal protective equipment [6,7]. Working as a nurse and being a woman were associated with greater mental distress in some studies [4,8], but not in others [3]. Staff were also reported to experience moral injury resulting from having to work in under-resourced services and providing suboptimal treatment [9].

HSCWs were already under considerable strain prior to the COVID-19 pandemic, with a growing incidence of stress, burnout, depression, suicide and substance misuse found across health occupational groups worldwide [10]. While HSCWs may experience similar work-related stressors, how they deal with them can differ. According to Lazarus and Folkman's [11] transactional model of stress and coping, stress is understood as a product of the interaction between an individual and their environment. This relationship is mediated by cognitive appraisals and coping. The cognitive appraisal process involves two parts; firstly, the appraisal of how threatening a situation is, followed by the appraisal of one's own ability to cope with it, which is based on the perceived internal and external resources an individual has available to them. Resources can be physiological, psychological, social or material. An individual will feel stressed if they perceive the demands of a situation as exceeding their resources for coping with it. On the other hand, an individual will not feel stressed if they perceive the demands of a situation as low and their ability to cope as high. These cognitive appraisals are followed by coping, which Folkman and Lazarus [12] defined as thoughts and behaviours individuals use to manage the internal and external demands created by a stressful situation. Individuals can cope and respond to stressors in diverse ways. Studies have found that individual differences also influence how people appraise and cope with stressful situations [13,14]. Therefore, many

variables can influence the coping experiences of HSCWs. Factors that have previously been found to contribute to burnout in HSCWs included demographic variables, personality characteristics, job characteristics, organisational variables and exposure to traumatic events [15–17]. However, so far most research has focused on risk factors for adverse outcomes at particular times in the pandemic, and usually during the early waves. Little is currently understood about how HSCWs' coping may have changed or developed over time.

The Covid-19 pandemic has highlighted the need to better understand how HSCWs cope with work-related stressors and what support would be most effective in helping them [18]. There is also a lack of evidence that directly incorporates the views and preferences of HSCWs themselves. Though important, this type of research does not help to understand the complexities of HSCWs experiences and does not take their views into account. Most published research on HCWS to date has been largely quantitative, focused on the early waves of the Covid-19 pandemic, has involved small samples, has been of poor to moderate quality, has been limited to doctors and nurses, and has only considered workers' experiences at one snapshot in time. [18]. However, coping is a dynamic and adaptive process [19]. Therefore, there is a need to understand how HSCWs have coped with this crisis over time.

The current study addressed these gaps by exploring the views of all types of HSCWs from 12–18 months after the peak of the first wave of the pandemic to understand what helped and hindered their ability to cope while working during the Covid-19 pandemic. A better understanding of the underlying factors that impact the coping experiences of HSCWs working in a pandemic can aid in the development of more appropriate evidence-based support for this occupational group. The aim of the study was to develop an explanatory model of the processes that helped and hindered the coping experiences of HSCWs and to illustrate how the factors that impact coping inter-relate, and in turn relate to help seeking.

## Method

### Study design, participants and procedure

The current study adopted an explorative qualitative research design using grounded theory methodology. Grounded theory employs inductive reasoning to generate a theory that is grounded in the data [20]. Grounded theory was considered an appropriate fit for the research aims of this study as through its focus on reciprocal effects between social processes and individuals it enables the in-depth investigation of the impact of social situations on patterns of behaviour [21], facilitating the development of a theory to explain behaviour [22]. The current study focused on understanding how social situations impacted on the coping behaviours of HSCWs. The resulting model was grounded in the data allowing for suggestions of causal connections and a greater understanding of how different factors inter-relate to either help or hinder coping in this population, providing an explanation in addition to identifying themes.

Frontline HSCWs were purposively recruited through social media (Facebook and Twitter) and the COVID Trauma Response Working Group website, and by snowball sampling through health and social care colleagues, asking them to share information about the study with potential participants. Any frontline HSCWs from a variety of disciplines who worked in the UK throughout the COVID-19 pandemic were invited to take part. Potential participants were excluded if they did not work in a frontline clinical or non-clinical role in the UK and did not speak English proficiently. A diverse range of participants from different professional groups, career stages and geographical locations was deliberately sought to access a wide range of experiences and views.

Individuals interested in taking part were invited to contact the first author by email. They were then screened for eligibility and sent the Participant Information Sheet and Consent

Form via reply email. Informed consent was obtained from all participants prior to taking part in the interviews.

Interviews were arranged at a convenient date and time for the frontline worker and took place by telephone or online video call. The semi-structure interview guide (see S1 File) was drafted collaboratively by the research team, in consultation with clinical and academic trauma experts from the COVID Trauma Response Working Group, and two frontline HSCWs, one working within a healthcare setting and the other working within a social care setting. Most interviews were conducted by the first author who received training on qualitative methodologies as part of her doctorate in clinical psychology and supervision from the last author to ensure methodological rigor. CT and ZK conducted one interview each, also under the supervision of the last author, and both received training on qualitative methodologies as part of their masters in clinical mental health sciences.

All interviews were audio recorded and transcribed verbatim. All potentially identifying information about the individual and their place of work were removed from transcripts to protect participants' anonymity. The interview recordings were deleted after transcribing was completed and checked. Numerical pseudonyms are used for illustrative quotes. All data collection and analysis methods complied with the terms and conditions agreed with participants and the ethical approval of this study.

Traditionally the concept of 'data saturation' is often used in grounded theory to determine sample size [23]. However, the use of saturation has been criticised in qualitative epistemology due to the variability in how it can be conceptualised and inconsistencies in how it is used [24]. The current study instead used the 'information power' approach to guide the decision about how many participants to interview [25]. Using the 'information power' approach, the sample size was determined by considering the study aims, sample specificity, use of established theory, quality of dialogue and analysis strategy [25]. This approach stipulates that the more the sample holds information that is actually relevant for the study, the less the number of participants needed. The information power approach was favoured and adopted in this study as the aim of the study was to specifically capture the underlying factors that helped and hindered coping rather than the entirety of participants' experiences of working in the frontline. All participating HSCWs would have had varying experiences due to differences in profession, personal life, health, age and work environment, which would have made data saturation difficult to attain. Therefore, by adopting the information power approach, recruitment continued until enough information on participants' experiences of coping was gathered. A sample size that offered a pragmatic balance between depth and breadth was sought [26].

### Ethical considerations

The study was granted dual ethical approval from the University College London Research Ethics Committee (Ref. 18341/001) and the Royal Holloway University of London Research Ethics Committee (REC Project ID: 2636).

During the interviews, participants were asked to discuss and reflect on potentially distressing past experiences. If participants reported significant distress, they were signposted to sources of psychological support. The research team were supported by regular supervision from the last author.

### Analysis

The analysis was conducted in accordance with grounded theory methodology as outlined by Charmaz [27]. In keeping with grounded theory, a constant comparative approach was used and data analysis commenced as soon as the first interview was transcribed. Data collection

and analysis occurred simultaneously throughout. All coding was conducted by the first author who sought immersion in the data by reading and re-reading all transcripts, reflecting on the interviews and writing memos.

The coding process within grounded theory has three stages: initial, focused and theoretical coding. During the initial coding stage, the transcribed interviews were coded sentence-by-sentence to facilitate detailed exploration of the data [27]. The last author reviewed and verified the initial coding of four interviews by the first author. As data collection and analysis were happening simultaneously, reflections after initial coding were written in memos, which informed later interviews and analysis.

All transcripts were imported into Nvivo Pro V12 for the focused coding stage. The most significant and frequent codes identified during initial coding were selected and used to analyse larger segments of data [28]. Codes were selected based on whether they made the most analytic sense to categorise the data thoroughly and completely. Using focused codes helped in exploring the reoccurring codes prevalent in the data and facilitated the linking together of initial codes to form concepts. The provisional coding frame was further extended and edited with the coding of subsequent transcripts. The last author reviewed the focused codes and provided feedback.

During the theoretical coding stage, the focused codes were reviewed together with the analytical memos to understand the relationships between them and establish how they can be unified into a theory. The focused codes shaped the theoretical codes and the relationships identified between the codes resulted in an emergent theory. Continuously writing memos helped in clearly defining the meaning of the codes and in organising and interpreting the relationships between them [29]. The theoretical and focused codes were then reviewed by the last author before a final visual representation of the model was developed to show the relationships between the key theoretical and focused codes. The draft model was then presented to academic peers conducting research in the same area, as a form of validity check. Modifications of the model were then made based on feedback received from this peer-group and from the last author.

The analysis of this qualitative data was rigorous, with all steps taken to maximise the validity and trustworthiness of the findings. Study rigour was ensured by adhering to the Standards of Reporting Qualitative Research (SRQR) outlined by O'Brien and colleagues [30]. The SRQR is a list of 21 items that are considered essential for complete and transparent reporting of qualitative research. To further ensure validity, health and social care workers with lived experience and clinical and academic trauma experts' perspectives were included in the design, delivery and analysis of this study.

## Reflexivity

The first author is a 32-year-old Arab heterosexual female trainee clinical psychologist, with an interest in post-traumatic stress disorder and trauma in occupational groups. The first author continued to work in client-facing roles throughout the Covid-19 pandemic in different settings within the NHS in London while the study was being conducted. Consideration of the first author's personal background highlighted differences and similarities between this author and participants. While the first author's experiences may have increased their understanding of the difficulties the HSCWs faced, it may also have impacted their objectivity. In order to enhance the credibility of findings, it was imperative to bring awareness to any preconceptions that may originate from the first author's point of view as the researcher conducting the data analysis [28].

CT and ZK are both MSc graduates in Clinical Mental Health Sciences. CB is an Advanced Nurse Practitioner with experience working in the Covid-19 frontline. DL is a Professor of

Neuropsychology at Royal Holloway, University of London. JB is a Consultant Clinical Psychologist and Associate Clinical Professor with over 20 years of experience of working in the NHS. She has particular expertise in the mental health and wellbeing of high-risk occupational groups. The research team was an ethnically diverse group, including White British, Arab and Asian backgrounds. The team members brought a range of different perspectives and experiences to this topic.

## Results

Twenty participants were recruited and took part in the study. The interviews took place between 5 May and 26 October 2021. They started from one year after the peak of the first wave of the pandemic in the UK. Interviews lasted between 37 minutes and 1 hour and 15 minutes, although most interviews took between 45 and 60 minutes. See Table 1 for participants' sociodemographic information.

Through our analysis of the data we identified eleven theoretical codes supported by specific focused codes, which are discussed below. Quotes from participants have been included

**Table 1. Participant sociodemographic information.**

| Gender | |
|---|---|
| Female | 18 |
| Male | 2 |
| **Role** | |
| Doctor | 3 |
| (Junior) | (1) |
| (Consultant) | (2) |
| Nurse | 9 |
| Care assistant | 1 |
| Physiotherapist | 1 |
| Clinical Psychologist | 2 |
| Midwife | 1 |
| Operating Department Practitioner | 2 |
| **Setting*** | |
| Intensive Care Unit (ICU) | 4 |
| Paediatric Intensive Care Unit | 1 |
| Paediatric ward | 1 |
| Accident & Emergency (A&E) Department | 3 |
| General hospital/COVID wards | 7 |
| Theatres | 2 |
| Acute ward | 2 |
| Outpatient Department | 1 |
| Community setting | 3 |
| Mental Health community | 2 |
| Care home | 1 |
| Psychiatric inpatient setting | 1 |
| Hospital-based antenatal clinic | 1 |
| Research | 1 |
| **Geographical location by UK region** | |
| London | 4 |
| South East England | 7 |
| South Central England | 1 |
| South West England | 2 |
| Midlands/Central England | 2 |
| North East England | 3 |
| North West England | 1 |

*Several participants worked across more than one setting during the first year of the pandemic.

for each of the focused codes to demonstrate how the codes are grounded in the data. A visual diagrammatic representation of the relationships between all the theoretical and focused codes is presented and discussed.

### Theoretical model of the factors that helped and hindered coping experiences of health and social care professionals working in the frontlines during the Covid-19 pandemic

The current study aimed to develop a theoretical model of the underlying processes that help and hinder coping in HSCWs working in the frontlines during a pandemic. Fig 1 outlines how the eleven theoretical codes relate to each other in an explanatory model. The blue lines represent the relationship between the theoretical codes and their respective focused codes, while the blue arrows present the processes described by participants of how certain focused codes relate and interact with each other. The temporal factors are represented with the orange arrow at the bottom of the figure to highlight the passing of the time throughout the first year of the pandemic and the differences in participants' experiences between the various time periods. These temporal changes occurred in parallel to the factors identified in the rest of the model. Personal context is presented by an oval beneath the rest of the model as participants stated that this had a significant and overarching effect on their ability to cope. Therefore, it forms the foundation of the model. Codes are presented starting with the system-related factors on the left and moving towards the more individual-related factors on the right. Systemic factors included resources, organisational response, management and colleagues, while individual factors included decision making and responsibilities, external impactors and internal impactors. Safety is represented as an outcome of all the other factors which fed into it as

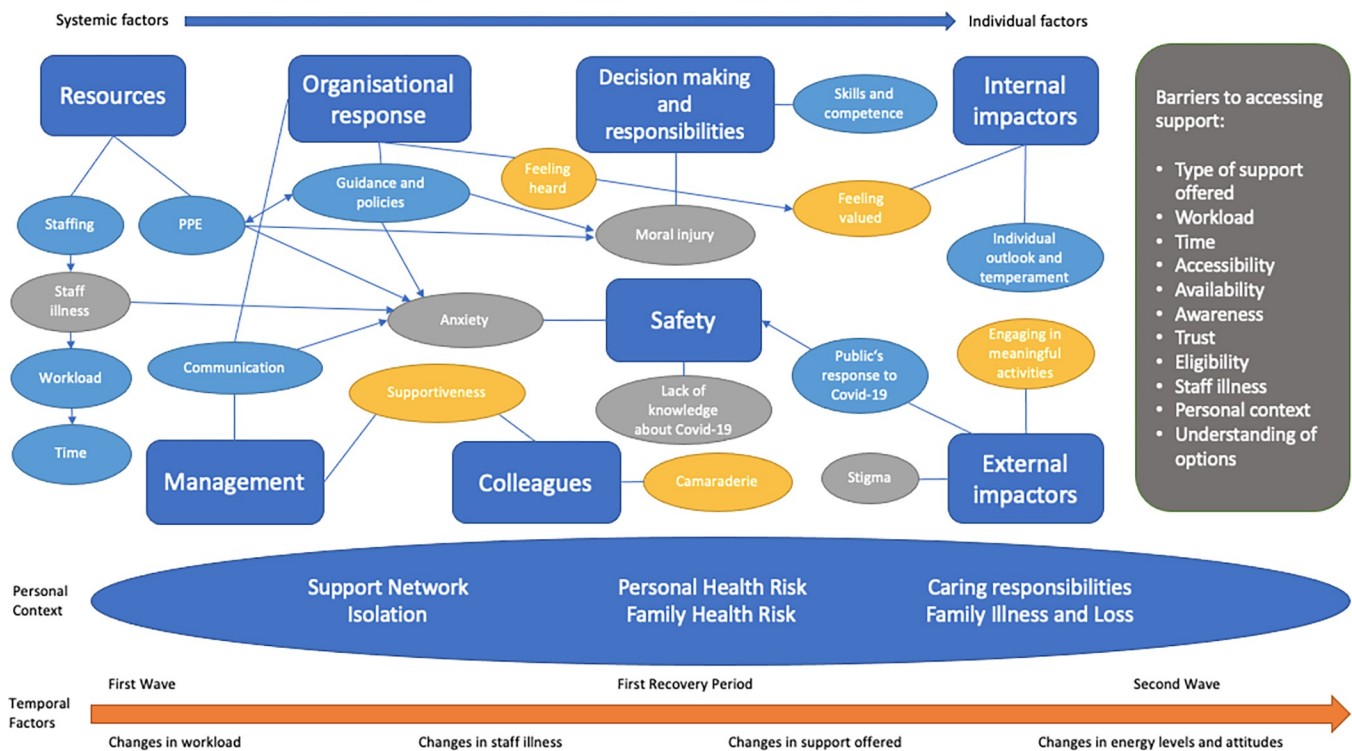

**Fig 1. Theoretical model of the factors that helped and hindered coping experiences of health and social care professionals working during the Covid-19 pandemic.**

participants stated that feelings of safety were integral to coping. The focused codes with a clear positive impact on coping are presented in yellow while the focused codes with a clear negative impact on coping are presented in light grey. The focused codes that are more neutrally worded are presented in light blue. The relationships between the various focused codes were represented in the model. For example, the focused code of PPE within the resources theoretical code had a direct influence on anxiety and moral injury. Anxiety was related to participants' sense of safety while moral injury was related to difficult decisions they had to make while working. PPE had a bidirectional relationship with guidance and policies because although guidance is meant to influence how PPE is used, many participants spoke about how during the pandemic guidance would change depending on PPE supplies, resulting in greater anxiety. Staffing was a resource that had a significant impact on coping as staff illness during the pandemic resulted in increased workloads and less time available. Guidance and policies set by the organisation influenced the anxiety and moral injury participants experienced. Feeling heard by the organisation impacted how valued participants felt. Feeling valued was an internal impactor from within oneself that affected participants' ability to cope. Communication from the organisation and management had a direct impact on anxiety. Supportiveness from both management and colleagues had a significant impact on coping. The public taking Covid-19 seriously had a direct influence on participants sense of safety. This was an external impactor, a factor outside of oneself, that affected participants' ability to cope. Other focused codes, such as camaraderie, stigma, engaging in meaningful activities and skills and competence, were included because they were highlighted by participants as significant factors that impacted their ability to cope.

The barriers for accessing support were presented because participants reflected on how even if they wanted to access support to help them to cope, there were various factors that hindered their ability to seek help. Factors impacting help seeking were discussed as distinct from factors that directly impacted coping experiences, therefore they were presented separately.

### Personal context

**Support network.**   Participants reflected on the importance of informal support provided by their personal support networks and how not being able to access this type of support at certain periods negatively impacted their ability to cope. Support networks had provided vital practical support by helping with caring responsibilities and preparing meals. Not being able to access this crucial support, particularly in relation to childcare, was particularly challenging.

*". . . it had been months and months and months of no one being able to help me or give me any support at all in terms of childcare" P2*

**Isolation.**   Participants spoke about feeling isolated from others both physically and mentally and how this negatively impacted on their ability to cope. They did not see family or friends for long periods of time. Many felt the need to isolate themselves from others because of their regular contact with Covid-positive patients as they worried about infecting others especially during the early phases of the pandemic before vaccines became widely available.

*". . . I felt like I couldn't or I didn't want to meet too many people outside of work with the risk of spreading because I was already meeting so many people who had Covid." P10*

**Family illness and loss.**   Participants reported how having to continue working while a family member was severely ill impacted their ability to cope. They described feeling more

worried and stressed, particularly when they were involved in the care of this family member or had to make difficult decisions regarding their care.

*". . . my dad got taken into hospital. [. . .] I remember going into work knowing that he's got COVID [. . .]. My mind was just, I couldn't concentrate. And I think I got a minute with no patients and I just broke down. . ." P19*

One participant spoke about how their partner died from Covid-19 which significantly impacted her ability to cope and resulted in her having to take a substantial amount of leave from work.

*". . . my well-being is at rock bottom because [. . .] my husband caught COVID, he brought it home and unfortunately, he died in May. [. . .] I'm still not able to go back to work because of the bereavement. . ." P11*

**Caring responsibilities.**   Participants reflected on the difficulty of having to juggle caring responsibilities for children and elderly relatives while working during the pandemic. As schools and nurseries closed, children had to stay at home and frontline workers could not access the support of their network due to lockdown. Participants spoke about how it was more difficult organising childcare with a partner that had to physically attend work and easier when their partner worked from home.

Single parents found it more difficult to organise childcare when the usual options were no longer available, with associated financial repercussions.

*". . . my little boy's nursery shut and with very little notice and I'm a single parent so I don't have anyone else at home and to be able to look after him [. . .] I was going to have to take unpaid leave so it was, financially very stressful as well. . ." P2*

Participants spoke about continuing to support their elderly parents with necessities such as food shopping. They worried about infecting them and spoke about how they noticed a deterioration in their parents' mental health due to isolation.

**Personal health risk.**   Participants reflected on how their level of personal health risk impacted on their ability to cope. Being considered medically low-risk made participants feel less worried about their safety while working and even made some participants want to volunteer for riskier tasks.

One participant spoke about how she was "*classed as at risk*" because she has "*asthma*" and how this made her feel "*frightened*" because she knew "*what that struggle is for breath*" P19, which made it difficult for her to cope.

Being pregnant made participants feel more anxious initially due to the uncertainty about how Covid impacted pregnancy and later on because they were prohibited from taking the vaccine.

**Family health risk.**   Participants reflected on how they worried about putting their family's health at risk. One participant commented on how not living with anyone "*vulnerable at home*" made her feel less "*concerned*" P8, while another commented on how having children at home made her feel more "*vulnerable*" P10. Some participants who had children that were medically vulnerable alleviated their anxiety by moving out of their home. Participants who stayed home introduced infection control measures for when they returned home and socialised their children to comply with them.

*". . . once I'd showered and cleaned and scrubbed, then I could be kissed and hugged." P14*

### Resources

**Staffing.**   Participants reflected on how staffing levels impacted their ability to cope, particularly when they were short-staffed, due to colleagues being on leave or shielding, as this resulted in increased workloads that were difficult to manage. Being redeployed to an unfamiliar environment also made it difficult to cope. One participant spoke about how *"there was nobody"* to give her an orientation so she *"had to hit the ground running and pick it up as I went along" P2*.

Being severely under-staffed increased participant's workloads and made them feel unable to take breaks or leave. Many spoke about how their departments were already under-staffed pre-pandemic. Unmanageable workloads forced participants to start making decisions based on *"what's the least unsafe care I can provide?" P4*

*"Both teams I'm in at the moment, completely overworked, completely burnt out, not enough staff anyway, let alone now sort of post-pandemic." P9*

Participants reflected on how they did not feel like they had a choice but to continue working because this was part of their job as care providers. Participants noted colleagues have resigned because of working during the pandemic, which is making them even more short-staffed and forcing their departments to work at *"a really unsafe level" P15*, reducing morale even more.

**PPE.**   Participants reflected on how the availability of PPE in their services impacted on their ability to cope. One participant spoke about how she felt *"lucky"* to be working for an *"incredibly well-resourced"* trust because she did not experience any *"issues about access to PPE" P3*. Others spoke about how they were frightened due to the lack of PPE.

*"We were very, very, very scared. Especially the fact that we lacked PPE so much that was the biggest fear factor." P15*

Many participants spoke about how the lack of access to PPE affected their ability to do their job and resulted in many colleagues becoming ill.

*". . . we had to make a decision that we wouldn't go into any of the rooms until someone could provide us with PPE,. . ." P12*

Constantly changing guidance about what PPE they had to wear made participants feel less safe, negatively impacting their ability to cope. Participants questioned whether the constantly changing PPE guidance was evidence-based and spoke about how the discrepancies between the guidance given to different departments reduced trust in the guidance overall.

*"It just felt like when PPE stores were running out, then suddenly the rules would change and there didn't seem to be any consistency. And you could go between one ward and the other, and on one place you'd need a full respirator mask, whereas the bed next door, you wouldn't. And it didn't really make much sense." P17*

**Facilities.**   Participants reflected on how the lack of space and facilities impacted their ability to cope. Not having any space to go to for breaks or to eat meals negatively *"affected morale"*.

*". . . you ended up having your lunch on your lap in the locker room. . ." P7*

The lack of space made it difficult to isolate Covid-positive patients, which made participants feel guilty for potentially harming other patients.

*"People were waiting in the waiting room with COVID, with everyone else, and we couldn't do anything about it to isolate them. We felt like we were almost hurting our other patients because we couldn't protect them." P18*

The use of video-conferencing technology helped improve participants' ability to do their work by making attending meetings *"easier" P8*.

**Funds.** Participants spoke about how financial resources and changes in salary influenced their ability to cope. During the pandemic, the *"money was found much quicker than pre-pandemic" P1* when departments requested funds to implement changes.

Participants appreciated being paid overtime for additional time spent at work and reported this made them feel valued.

*". . . they paid us half an hour more for every shift we did because they know that it took longer to go in and go out because of all the PPE." P14*

However, participants were disappointed by the government's decision to cut the salary increase of healthcare workers resulting in them feeling undervalued.

*". . . getting a one percent salary increase has really felt like a slap in the face for many people." P14*

## Organisational response

**Guidance and policies.** Participants reflected on how continuous and frequent changes to guidance and policies impacted on their ability to cope. The lack of clear guidance resulted in feelings of frustration and distrust of the organisation as well as the individuals responsible for issuing the guidance. Participants felt nobody knew what they were doing, which made them feel less safe. Participants highlighted how PPE policies being issued by their trusts differed from those issued by their professional bodies, which made them question their validity.

*". . . it doesn't instil you with much confidence because a lot of the time, it didn't feel like the guidance, they knew what they were doing, so that made a lot of people feel quite unsafe." P11*

The lack of clarity in the way guidance was written meant it could be interpreted in different ways which made it difficult for participants to understand and resulted in conflict between team members. Participants felt more responsible for negative outcomes and experienced greater feelings of guilt due to the lack of clarity.

*". . . a lack of guidance made it really difficult to cope because it's like everything becomes really personal. It's like have I killed people?" P4*

**Feeling heard.** Many participants felt their voices and concerns were not taken into consideration or given importance by their organisations. The top-down communication made participants feel their organisation did not care to hear or understand what their experiences were like.

*". . . they were quite happy to just implement things and expect us to work with this, like face to face without proper PPE, and they weren't really listening to our concerns." P18*

Participants questioned how their organisation would be able to provide adequate support for them without trying to understand their challenges.

*"Listen to the frontline staff and then make changes based on what they say, rather than sitting in an office deciding what we're going to do. Making decisions that doesn't affect them at all but affects us massively." P7*

Participants who felt heard reflected on how this had a positive impact on their ability to cope because it made them feel valued and cared for.

*". . . just that act of listening and checking in and touching base and feeling like someone cares about who you are as a human being [. . .] makes a massive difference." P2*

**Communication.** Participants reflected on how the style and rate of communication from the organisation they worked in impacted their ability to cope. The lack of communication made participants feel more anxious.

*"Changes were starting to be made in preparation without much communication to the people actually working on the shop floor that made us more fearful because it was almost like they knew something we didn't." P18*

Participants that received regular communications from their organisation through executive briefings where they could also provide input felt this was helpful because it made them feel like the usual *"hierarchical barriers" P8* were absent. However, receiving daily communications about the rates of infection and death in the hospital were viewed as unhelpful as they made participants feel less safe. Participants preferred these to be optional. Participants said the *"the whole 'we're all in this together' pep talks from people that were not on the frontline" P7* were not helpful and resulted in feelings of resentment.

**Pace and management of changes.** Participants reflected on how they had to cope with ongoing changes that were being suddenly implemented and made their work more stressful.

*"It was all we're doing this today, but tomorrow we're doing this." P13*

Participants who were redeployed were told that their services were deemed *"non-essential"* and had their activities *"ceased almost immediately" P17*. Participants reflected on how having their work declared non-essential upset them.

Participants spoke about how their redeployments were also halted swiftly. They found it difficult immediately returning to their regular roles without a break. It was difficult returning to their teams because all colleagues had such different experiences. Some were redeployed, while others were not. Participants felt pressured to get their services back up and running quickly to cope with the backlog of patients.

*"We'd all sort of separated during COVID, some people had gone to intensive care. Some people had stayed at home in the garden. [. . .] It was a really odd experience to come back together as a team. And there was no recovery time because then we were under a lot of pressure to catch up on the waiting list. . ." P3*

**Level of preparation.** Participants reflected on the amount of preparation they were given before their roles and responsibilities were changed, and how this made them feel. Changes being implemented suddenly resulted in participants having little time to prepare for their new responsibilities. For example, participants who were redeployed reported they did not receive any training and were expected to "*hit the ground running" P2.*

*"I went back to work on the Tuesday and was sent straight into COVID ITU. There was no training, there was nothing." P7*

One participant who received *"in-depth ITU scenario-based training" P19* at the start of her redeployment, compared this experience to her previous experiences working through other outbreaks. The greater level of preparation for Covid-19 made her worry more because she felt it was being taken more seriously than others.

### Management

**Supportiveness.** Many participants experienced a lack of support from their managers which made them feel abandoned. Few participants highlighted how the lack of support from management was due to managers being off sick.

*". . . we felt we were just left to get on with it" P5*

The lack of support resulted in staff no longer having *"respect for that hierarchy"* because they *"felt so let down by them [management]" P17.* The lack of support from management was also described as dehumanising.

*"It felt very much like throughout the whole thing that NHS staff were just seen as this kind of massive machine and we were all just cogs that needed to keep turning in order for everything to keep working, like at whatever cost, and if you didn't have Covid you were meant to be there, being that cog and if you weren't being a cog, you were not useful to them and they were not interested in you and it just felt all very kind of unhuman." P2*

Participants who felt well supported by their managers said they felt safer because their managers listened to their needs and advocated for them.

**Communication.** The inconsistency and lack of communication from managers made participants feel more anxious. Participants said being informed by their managers about patient bereavements ahead of a shift was helpful and *"was a nice human touch" P19.*

*"I wish that I would have been told before coming on shift who died, [. . .] I would have appreciated that." P4*

From the perspective of a manager, one participant spoke about how she held regular meetings with different professionals within her team to listen *"to what people had to say" P6* and offer tailored support when needed.

**Understanding and acknowledging challenges.** Participants felt management did not understand or acknowledge the extent of their challenges which made it more difficult for their managers to support them. Every role had a unique set of challenges, which would have been helpful for management to know. Many participants suggested that management *"come and do a shift" P4*, to understand how to better support them.

Managers who communicated with participants in an understanding way helped them feel less stressed by alleviating the pressure.

*"What I think was very helpful was that while we were in ITU, they [managers] would say, "you can only do what you can do, don't beat yourself up about something. If it hasn't happened, just hand it over. The situation is uncontrollable and relentless". And so giving yourself permission". P14*

**Empathy.**  Participants experienced a lack of empathy from management and felt their concerns were dismissed. Participants found the lack of empathy particularly difficult when they were in redeployed roles because they were suddenly responsible for completing tasks they did not have the skills for. The lack of empathy resulted in *"a few people just walking off shift".*

*". . . no one really was concerned about your mental health or how you were feeling, and if you did voice a concern about how you're feeling, it was kind of brushed aside [. . .] It didn't matter that staff were going home in tears or couldn't sleep and stuff." P7*

Participants who were medically vulnerable and experienced personal difficulties spoke about how the lack of empathy from management upset them and made them feel like an *"assignment number" P19.* One participant spoke about how the lack of empathy from management stopped her being offered support.

**Visibility and availability.**  Participants spoke about how they did not see their managers and described them as *"non-existent" P17,* which made them feel unsupported and angry.

*"They just left us to it and they all went to work from home. We didn't see anybody management wise for over three months, so we all got very, very crossed." P5*

Some participants spoke about how they felt their managers were unavailable because they were *"spread so thin" P14* and too busy themselves.

Participants working on Covid wards reported they could not approach managers even if they wanted to, because of infection control protocols which made it difficult to reach them.

**Respecting staff time.**  Participants reflected on how management respecting their time facilitated coping. Participants who experienced managers respecting their time, by not expecting them to stay late and making sure they took breaks, found this helpful.

*"No one was expecting you to stay more to finish them, which is a mentality that is very present in ITU if it is not under those circumstances, so I think that was very helpful." P14*

Participants that were pressured by their managers to regularly stay late said this negatively impacted their ability to cope. One participant commented on how this resulted in colleagues going *"off with stress" P7.*

## Colleagues

**Supportiveness.**  Support from colleagues was described as vital in helping participants to cope.

*"The doctors, my colleagues, ODPs nurses, cleaners, domestic people, all very supportive. And, two-way things, you supported each other through it." P11*

Participants spoke about how the hierarchies that were usually in place disappeared during the pandemic as colleagues from every level were helping each other out and *"started working as a big globular group" P1.*

Participants spoke about how colleagues offered both practical and emotional support, which they appreciated and found helpful.

*". . . the night where I had quite a few patients who died, my colleague was able to be like, "Go on, break, I'll cover you" like to just make sure that you get those breaks and things. To make sure you get out of the PPE and that you get a drink and that kind of thing." P12*

*"They [colleagues] were an important kind of emotional support. . ." P8*

One participant commented on how her colleagues were initially supportive but over time, as the whole team became more exhausted and felt an increase in pressure, colleagues started *"getting a lot snappier with each other" P18.*

One participant, based in the community, reflected on how not physically going into work meant she could not receive informal support from colleagues.

**Camaraderie.**   The sense of camaraderie that developed with colleagues had a significantly positive impact on coping. One participant compared the camaraderie he experienced during Covid-19 to his time in the UK military.

*". . . things that made it easier was the sense of camaraderie between the team. We were like rallying together like we were all in this and we all understood what each other were going through." P12*

One participant, based in a community team, reflected on how the lack of camaraderie with her colleagues had a negative impact on her ability to cope.

**Relatability.**   Participants reflected on how talking to colleagues was helpful because they could relate to how they were feeling as they were *"very much in the same boat" P13.*

Learning that colleagues felt equally anxious helped participants to cope because it normalised their own feelings.

*". . . having other people who are going through the same thing and your feelings [. . .],made me feel a bit reassured that it's just fairly normal to feel so anxious during such unsafe times." P10*

Participants felt it was easier talking to colleagues rather than friends or family because they would not be able to relate to their experiences and may not want to hear about negative subjects such as death.

*". . . people don't really want to hear how that person died, [. . .] It's not always the sort of stuff you can talk through with your kids or wife because they don't really understand." P16*

**Burden of helping each other.**   Participants reflected on how they felt compelled to help their colleagues which felt burdensome at times. Participants helped their colleagues despite their reluctance because they felt guilty if they did not.

*"Sometimes you'd stay late or you try and kind of cover people, so you might not take a long break because you knew that your break then impacted on somebody else." P12*

**Caring for colleagues.** Participants reflected on how caring for their own colleagues impacted on their ability to cope. Participants reported feeling more anxious and less safe when they had to care for a colleague.

*"We had a lot of staff off sick and we were also looking after staff. [. . .] Looking after your colleagues is obviously not something that you ever want to do and makes you nervous." P12*

**Colleague bereavements.** Participants reflected on how difficult it was for them to cope with the loss of colleagues to Covid-19, particularly after they had to care for them.

*". . . when your colleagues have died from it and they're in hospital and they're unwell and you're getting text messages saying how scared they are, they feel like they can't breathe and you're just helpless, you really are just helpless." P19*

## Decision-making and responsibilities

**Skills and competence.** Not having the skills and competence to make certain decisions made it difficult for participants to cope with the demands of their role, especially those that were redeployed.

*"I had to make decisions that otherwise someone far more senior would make. It's like I'm unqualified. I don't know. . ." P4*

Participants relied on colleagues to help train and teach them the skills they lacked when needed.

*". . . it was all kind of clinician lead so if you had someone in your team who had a skill, they would teach you it. . ." P17*

**Moral injury.** Participants reflected on how they experienced moral injury because of difficult decisions they had to make while working during the pandemic. Participants found conversations with patients and their families particularly challenging.

*"I could not believe myself that I was having to say it to people: 'sorry it's been six years of marriage and your wife is not actually allowed to come in'." P1*

Participants spoke about how the lack of clear guidance or having to follow guidance they disagreed with, resulted in them making morally injurious decisions.

*". . . when you feel like you're being blamed for decisions other people have made and you can't really do anything about them, like it's not a nice feeling. You feel so guilty, . . ." P8*

Participants spoke about how the lack of PPE resulted in them having to make morally injurious decisions such as withholding interventions, which was difficult to cope with.

*". . . you no longer could do CPR without an FFP3 mask on, and we didn't have enough FFP3 masks to put one in every room. So if you found a patient unresponsive and you pulled the crash bell, you weren't allowed to do chest compressions until somebody got there in full PPE. We just had to stand there and that just goes against everything that we're taught." P12*

**Complexity.**   Participants reflected on how difficult it was to cope with the complexity of the decisions that had to be made.

*"For instance, you have people who need to use like a hoist and stuff, so you need two carers. But there weren't two carers in the building, so it's like do I unsafely move this person, or do I just keep them in bed all day? Neither are good. . ." P4*

*". . . there was one time where there was only one ICU bed left and I had three patients in resus that needed it." P18*

**Consequences.**   Participants spoke about how ruminating on the consequences of the difficult decisions they had to make, made their work more stressful. Thinking about how they could have taken the wrong decision made it more difficult for them to cope.

*"I just went home feeling like I'd given her a death sentence, like if she caught Covid because I hadn't isolated her. It would be my fault if anything happened to her." P18*

## External impactors

**Engaging in meaningful activities.**   Participants commented on how being able to engage in meaningful activities outside of work helped them to cope. Continuing to engage in activities that were *"non-medical" P8* was helpful for participants; such as creative pursuits, exercise and church.

*"I did quite a lot of crafting at the time just because it took my mind off it [work]." P7*

Some participants spoke about how feeling exhausted and burnt-out from work stopped them from being able to engage in meaningful activities in their limited time off.

*"I used to be really involved in like sea swimming and I'd spend a lot of time doing it. I was involved in a club and stuff, but then I was just so burnt out from work. I just didn't have the energy to do anything in the evening." P20*

**Public taking Covid-19 seriously.**   Seeing the public take Covid-19 safety regulations seriously helped participants to cope because it made them feel less worried about their safety. While seeing the public not take Covid-19 seriously made them feel less safe.

*". . . when the general public were wearing face masks. It just made you feel that little bit more reassured." P17*

Few participants reported witnessing the public not following Covid restrictions made them feel angry.

*"And I just want to scream at them, which is not a very healthy reaction" P14*

**Stigma.**   Participants commented on how they felt stigmatised for working as healthcare professionals during the pandemic because people did not feel safe around them. Participants were not invited to events by family and friends.

*"She's one of my best friends, but she was like, "I can't invite you because my sister said you're working with COVID people and she doesn't want you to be there and she won't come if you were there", because people were so worried that I was carrying it" P12*

One participant who was living in a flat-share with other healthcare workers spoke about how their landlord did not allow them renew their lease because of their jobs.

### Internal impactors

**Feeling valued.** Being supported by their organisations and the public made participants feel valued which helped them to cope.

*"I think my trust in particular has been very helpful. I know other hospitals haven't been as helpful. So we felt very supported, very valued." P14*

Participants who experienced a lack of support from their organisations did not feel valued.

*". . . there were a lot of people that were just angry, angry at the fact that we were just left, like, people really felt undervalued. . ." P15*

Participants commented on how not being recognised appropriately for their efforts made them feel undervalued by their organisations.

**Individual's outlook and temperament.** Participants reflected on how their own attitudes while working during the pandemic impacted on their ability to cope. Participants with an optimistic outlook on their experiences spoke about how this helped them to cope. One participant commented on how imagining a future where Covid is no longer a threat helped her to cope.

*"I think just having the mindset [. . .] that this is going to be over. [. . .] I think just kind of visualising a future where COVID isn't such a big problem. I think that that's what helps." P8*

Participants reported feeling *"bitter" P12* and *"a bit jealous" P9* of friends and family that were able to stay safe by working from home. Some participants spoke about how they noticed themselves becoming angrier.

**Feeling helpful.** Participants reflected on how feeling helpful during a crisis helped them to cope. One participant spoke about how being able to work during the pandemic made it easier to cope because it made her feel *"less powerless" P4*, while another participant spoke about how working in the frontlines made her feel *"proud" P8*.

*". . . being able to support the team as best I could and support patients and families as best I could. I found rewarding in a way" P3*

**Physically connecting with others.** Participants reflected on how they appreciated being able to physically go into work and connect with others during the pandemic, especially when compared to working from home which was more isolating.

*". . . it's just so much better being able to go in and see people and do things with your hands and like be out and about, than work from home." P4*

**Safety.** Participants reported their sense of safety was influenced by their level of anxiety, lack of knowledge about Covid-19, infection control procedures, testing and the vaccine.

Participants commented on how the **anxiety** they felt was fuelled by the lack of PPE, constantly changing PPE guidance and media reporting during the Covid-19 pandemic.

*"There was an obvious anxiety at first, because when we first started working, I don't think we had any PPE. And then it went from having nothing at all, not even your paper face masks, to then get them say "no actually you need a full respirator mask, hazmat suit" and overnight it changed and I think also on the ICU, we were wearing full PPE, but then we would go onto a ward and we didn't have anything. The disparity between it was quite concerning." P17*

Participants reflected on how the **lack of knowledge** about the novel virus and about how to treat it impacted their ability to cope. The uncertainty surrounding Covid-19 and how to treat it made participants feel more worried because they felt "*no one really knew what to do about it*" P17.

Participants reflected on how the **infection control procedures** implemented within their services made them feel safer. Participants who witnessed infection control procedures not being followed felt more anxious.

*"There was no social distancing at all in the office, didn't have to wear masks and all just in the same office, just all feeling quite uncertain [. . .] it was really anxiety provoking." P9*

Participants spoke about how being able to get **tested for Covid-19** and knowing their Covid status made them feel less anxious. Participants also reflected on how they felt safer after receiving the **vaccine** and after their colleagues received the vaccine.

## Temporal factors

**Changes in staff illness.** Participants reflected on how staff illness levels changed over time during the first year of the pandemic and the impact this had on them. Participants noted there was greater staff illness in the second wave, which resulted in increased workloads.

*"We were working ridiculous amount of overtime hours just because we were so short staffed. I think by the time the second wave came around like, I think at one point staff sickness was around 30%. . ." P17*

A participant who worked in a care home spoke about how in her service they experienced greater staff illness during the first wave and none during the second wave.

**Changes in workload.** Participants reflected on how their workload changed over time during the first year of the pandemic. Most participants spoke about how their workload increased during the second wave due to staff illness, increased severity of Covid variants and patients avoiding seeking help throughout the first wave. Only two participants, including the care home worker, said they just experienced an increased workload during the first wave.

*". . . there was a lot more people coming through the doors the second time around, in comparison to the first. Whereas the first time around people were a lot sicker, we had patients just piled up on the corridors waiting for beds, ambulances queued up outside the second time. . ." P17*

**Changes in energy levels and attitudes.** Participants reflected on how their energy levels and attitudes about working in the frontlines changed over time and how this had a negative impact on their wellbeing. Participants reported that the second wave was more difficult to

cope with because by that point the novelty of the pandemic had worn off and they were feeling exhausted from the *"relentlessness of it"* P3.

*"Everyone found it a lot tougher the second time around [. . .] first time around it was new, it was different, it was a challenge, it was a bit of problem-solving, whereas the second time around, it just felt like you were fighting a losing battle and everyone was exhausted, everyone was really stressed."* P17

**Changes in support offered.** Participants reflected on how the support they were offered changed over time during the first year of the pandemic. Some participants spoke about how they were offered more support during the second wave, due to management focusing on staffing issues and managing general *"panic"*.

*"The first wave, everyone was just reeling and just completely overwhelmed and I think they tried to basically make sure there was enough staff everywhere. [. . .] and then the more emotional support stuff came a bit later when I think they realised how long it was going to go on for, what a big impact it was having on people and management had time to get their head around things and actually sort that out."* P2

While other participants spoke about how they were offered more support during the first wave as greater efforts were made to alleviate distress during the onset of the crisis.

*". . . it was within the first wave, I was most aware of it [support offered], [. . .] I've been less aware of it during the second wave."* P3

## Barriers to accessing support

Participants spoke about how their **workloads** stopped them from being able to access support.

*"I think just work pressure in general and workload was a massive barrier to accessing support."* P17

Participants spoke about how the **type of support offered** was a barrier for them accessing support. They did not view the support being offered as helpful.

*"There are things [support offered] I've avoided doing because I've looked at it and gone, 'oh my goodness, there is no way that is a good idea. I'm just not going to go and do that'"* P3

Participants reflected on how **time** was a barrier to accessing support. Participants did not have enough time to access support due to their increased workloads. There was also the belief that support would have to be accessed during their own personal time, which they did not want to do.

*"It was like manic. You didn't have time to think of yourself or, it was just keep on going, it wouldn't have even come to my mind to get support"* P11

Participants commented on how lack of **awareness** of what support was available was a barrier to accessing support.

*"They [support offered] weren't particularly well advertised. . ."* P7

Participants spoke about how lack of **accessibility** was a barrier to accessing support.

*"I've got loads of stuff to do, so they needed to put stuff in place so that it was like 'this person is going to do all of your jobs for the next 20 minutes. We are going to come and talk about, Are you OK? What do you need? What would be helpful?'" P12*

Participants reflected on how the lack of **availability** of support was a barrier to accessing support. A mental health worker reflected on how she provided support but was not offered any. A care worker commented on how she *"got absolutely no support" P4* offered to her.

Participants spoke about how **staff illness** was a barrier to accessing support because it resulted in increased workloads and less time available.

*". . .just the sheer amount that I was working, just doing all the overtime that we could because we were so short staffed, [. . .]. So that made it harder to access anything." P18*

Participants commented on how **trust** was a barrier to accessing support because they did not feel the support offered by the organisation would be confidential or *"authentic" P15.*

*". . . whilst they do say that it's all confidential and things like that, you do wonder how confidential it is and if it will get back to someone. . . ." P13*

Participants spoke about how a lack of **understanding of the different options** available and of what type of support might be the most helpful for them was a barrier to accessing support.

*". . . it was difficult to know what to access. [. . .] I know that other people are also very confused by what it is they should be accessing." P3*

Participants reflected on how their **personal context** and responsibilities outside of work were a barrier to accessing support.

*". . .it [accessing support] would have had to have been outside of my working hours, which I would have never been able to do because I didn't have childcare. . ." P2*

Participants spoke about how not being **eligible** was a barrier to accessing support, particularly for the social care worker.

*". . . order a pizza and it was like NHS staff get a quarter off and it's like can I get a quarter off? It felt like care workers had just completely been forgotten about." P4*

## Discussion

The current study explored the underlying factors that impacted the coping experiences of HSCWs working in the frontline during the first year of the Covid-19 pandemic. The study aimed to develop an explanatory model of the processes that helped and hindered the coping experiences of this occupational group and illustrate how they inter-relate. The results highlighted that HSCWs' coping experiences were complex and distinct.

Many HSCWs were unable to work in their usual roles either due to illness, shielding or being redeployed which greatly reduced staffing levels and increased workloads for those that continued to work. Increased workloads made it more difficult for HSCWs to take sufficient

breaks which they reported was further exacerbated by the lack of facilities available for them to rest, in line with other findings [31,32]. Most HSCWs identified lack of access to PPE resources as a major factor that hindered coping because it increased anxiety due to concerns about safety and ability to do their job. Considering the transactional model of stress and coping, the lack of physical resources, such as PPE, resulted in HSCWs appraising that they lacked the ability to cope with the work-related stressors they were facing, such as not being able to do their job, resulting in greater stress [11].

HSCWs identified lack of clear and consistent guidance and not feeling heard by their organisation as significant factors that hindered their ability to cope. Continuously changing guidance led to mistrust as many speculated that changes to guidance were based on changes in PPE supplies available within their organisation rather than the evidence-base around what protection was most effective. Typically it is the guidance, based on the evidence-base, that dictates what PPE should be used, not vice versa [33]. The lack of clear guidance results in increased anxiety as HSCWs worry about their safety [34,35]. The lack of clear guidance and PPE supplies also resulted in experiences of moral injury because HSCWs described not being able to provide the care they felt morally obliged to provide. Moral injury has been highlighted in the literature as a significant concern for HSCWs during Covid-19 [36]. Many HSCWs reflected on how not feeling heard by their organisation made them feel undervalued. They questioned how their organisation would be able to support them without asking them directly about their experiences to understand the unique challenges they were facing. HSCWs desired two-way, consistent and regular communication from their organisation.

Management had a significant impact on HSCWs experiences of coping. The lack of support and communication from management hindered coping because it increased their anxiety. HSCWs believed managers did not fully understand the challenges they were facing and wondered how they could without regular communication and being physically present in the workplace environment. Receiving regular appropriate information from management during the Covid-19 pandemic has been found to help HSCWs to better cope [37]. Many HSCWs said they experienced a lack of empathy from management, particularly regarding their personal health risks or personal context. Not being able to tell managers when they are not coping has been found to negatively impact HSCWs wellbeing [3].

HSCWs valued supportive and compassionate relationships with colleagues. Support from colleagues greatly facilitated coping and HSCWs stated they valued this support over any other because colleagues could relate to their experiences as they were "in the same boat". In addition to supportiveness, the increased sense of camaraderie with colleagues helped facilitate coping as they rallied together while working in the frontline. Camaraderie has been identified as a protective factor in other research [36]. As in the literature, the camaraderie experienced was compared to working in a military frontline [38]. However, at times HSCWs felt burdened by colleagues. Particularly when helping them involved taking on additional shifts and less breaks. In the literature, HSCWs described feeling burdened by colleagues when they had to support those with emotional difficulties as they worried about offering appropriate advice [39,40].

Feeling valued, whether by their organisation or the public, was a key factor that had a positive impact on HSCWs coping experiences. Similarly to findings from previous pandemics [37,41], not feeling heard by their organisation made HSCWs feel undervalued which negatively impacted their ability to cope. HSCWs appreciated being recognised appropriately for their efforts. Support from the public helped boost morale and facilitated coping because it made workers feel valued. This is mirrored in other studies conducted during Covid-19 which found that support from the public resulted in HSCWs feeling appreciated and empowered by their communities [42,43]. However, witnessing the public disregarding Covid-19 regulations

hindered coping by making HSCWs feel less safe and more angry, which was echoed in the literature [44]. Stigmatisation and fear from others for being a HSCW was also widespread during Covid-19 [3,45,46], leading to further isolation.

As HSCWs reflected on their journeys throughout the first year of the Covid-19 pandemic, certain factors changed over time which impacted their ability to cope. All healthcare workers stated that staff illness was greater during the second wave and most reported this resulted in greater workloads during this period. While a care worker explained that in the care home they experienced one wave only, therefore staff illness and workload were greater during that period. Experiences of support offered varied between services. Some HSCWs were offered more support during the first wave while others were offered more support during the second wave as management had more time to organise this than during the onset of the crisis. Billings and colleagues [39] found that having support offered and then taken away made HSCWs feel undervalued. HSCWs would appreciate consistency in support offered. With regards to energy levels and attitudes, the novelty of the pandemic during the first wave was met with greater energy as managing the crisis was perceived as a challenge. However, by the second wave the novelty had worn off and HSCWs described feeling exhausted from the relentlessness of their workloads. Individual differences in a HSCWs' outlook and temperament were found to impact their ability to cope. Being optimistic and future-orientated were found to facilitate coping, while experiencing feelings of bitterness and anger hindered coping. Research has shown that positive emotions play a crucial role in enhancing coping resources for individuals experiencing negative events [47].

HSCWs recognised that they struggled to cope at times and spoke about the support they were offered but reflected on the many barriers to accessing support. By offering different forms of support to HSCWs, organisations will increase the perceived resources a staff member has available to them when facing a stressor, which will make them feel more able to cope [11]. There was a striking variety of experiences amongst the HSCWs, with some reporting that they had not been offered any form of support by their organisation. Even when support was offered, some reported that there was a lack of awareness about available support and perceived that it was not easily accessible. HSCWs highlighted work-related barriers such as greater staff illness and increased workloads made it difficult to access support during working hours. HSCWs assumed that they would have to access support during their non-working hours which they did not want to do as personal context and family responsibilities were considered another barrier to accessing support. This finding is echoed in another study where staff explained how support was usually offered during working hours which made it difficult to access because they did not have the time during the workday to attend [39]. The importance of making support easily accessible for staff was echoed in other studies [48,49]. There were barriers directly related to the support offered. HSCWs did not always understand the various options available, particularly when there were many, and described feeling overwhelmed by them. Some HSCWs, such as mental health professionals, were not aware that they were eligible for the support being offered. While social care workers who were not employed by the NHS were unable to access a great deal of support being offered by organisations and the public because they were not eligible. The findings of this study suggest that organisations would benefit from making support easily accessible and equitable for all staff.

## Strengths and limitations

The results of the current study should be considered within the context of its strengths and limitations. Although there has been a surge in research focusing on the experiences of HSCWs during the Covid-19 pandemic, to the knowledge of the research team none have

focused on developing an explanatory model outlining the individual and social processes that influence their coping experiences. There is also a lack of research which focuses on the changes over time of the experiences of frontline workers during Covid-19. Thus, the current study addressed these research gaps. The analysis of the qualitative data was rigorous, with all steps taken to maximise validity. Health and social care workers and clinical and academic trauma experts' perspectives were included in the design, delivery and analysis of this study. The sample was diverse in terms of the settings participants worked in which suggests the current proposed model could be transferrable to a wide range of health and social care settings. Another strength was conducting the interviews from one year after the first peak of the first wave as this gave HSCWs ample time to reflect on their experiences and understand what helped and did not help them to cope. This also highlighted how HSCWs needs changed over time.

Nonetheless, this study has a number of limitations. There is a lack of diversity in the sample in terms of profession, gender, ethnicity and regions worked in. As the findings of the current study are only related to UK-based health and social care workers, the results are further limited in their transferability to HSCWs based in other countries experiencing a different context.

## Conclusion

The current study provides an in-depth analysis of the factors that facilitated and hindered the coping of frontline HSCWs during the first year of Covid-19 and illustrates how they interrelate and impact help-seeking by developing an explanatory model. The results of this study show that there are numerous factors related to the system as a whole and the individual which can impact on coping. Therefore, a "one-size fits all" approach to offering support would be unhelpful. It is hoped that by identifying the various factors that impact on HSCWs coping experiences and help-seeking behaviours, strategies can be developed and implemented to better support this occupational group when facing future health crises.

## Supporting information

**S1 File. Interview guide.** This is the interview schedule which was used to guide the semi-structured interviews.
(XLSX)

## Acknowledgments

We would like to thank all the frontline health and social care workers who took part in this research. We would also like to thank the COVID Trauma Response Working Group, UCL Trauma Research Group and our health and social care colleagues who provided valuable guidance on the design, delivery and analysis of this study.

## Author Contributions

**Conceptualization:** Karina Soubra, Camilla Tamworth, Zeast Kamal, Clare Brook, Dawn Langdon, Jo Billings.

**Data curation:** Karina Soubra, Camilla Tamworth, Zeast Kamal.

**Formal analysis:** Karina Soubra, Camilla Tamworth, Zeast Kamal, Jo Billings.

**Methodology:** Karina Soubra, Jo Billings.

**Project administration:** Karina Soubra.

**Supervision:** Dawn Langdon, Jo Billings.

**Validation:** Camilla Tamworth, Zeast Kamal, Clare Brook, Dawn Langdon, Jo Billings.

**Writing – original draft:** Karina Soubra.

**Writing – review & editing:** Jo Billings.

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
