## [Decision Letter · Decision Letter 0]

13 Feb 2023

PONE-D-22-34685Health and social care workers experiences of coping while working in the frontline during the COVID-19 pandemic: one year on.PLOS ONE

Dear Dr. Billings,

Thank you for submitting your manuscript to PLOS ONE. After careful consideration, we feel that it has merit but does not fully meet PLOS ONE’s publication criteria as it currently stands. Therefore, we invite you to submit a revised version of the manuscript that addresses the points raised during the review process.

We look forward to receiving your revised manuscript.

Kind regards,

Muhammad Arsyad Subu, Ph.D

Academic Editor

PLOS ONE

Journal Requirements:

2. In your Methods section, please include additional information about your dataset and ensure that you have included a statement specifying whether the collection and analysis method complied with the terms and conditions for the source of the data.

Reviewers' comments:

Reviewer's Responses to Questions

**Comments to the Author**

1. Is the manuscript technically sound, and do the data support the conclusions?

Reviewer #1: Yes

2. Has the statistical analysis been performed appropriately and rigorously? 

Reviewer #1: N/A

3. Have the authors made all data underlying the findings in their manuscript fully available?

Reviewer #1: No

4. Is the manuscript presented in an intelligible fashion and written in standard English?

Reviewer #1: Yes

5. Review Comments to the Author

Reviewer #1: Thank you so much for the opportunity to review this interesting manuscript. Please see my comments and suggestions below:

1. Abstract

Comment: Please follow the guidelines in writing your abstract.

2. Interviewer/facilitator

Comment: Which author/s conducted the data collection? What experience or training did the researcher have?

3. Study design

Comment:

• Research Ethics Committee (Ref. 18341/001) in page 6 should be put in ethical considerations section (page 7).

• Grounded theory as the methodological orientation must be stated in this study design section. Please describe this design (Charmaz constructivist GT). I suggest authors to provide a brief explanation of research paradigm or worldview of this adopted GT.

Participant selection and setting

Comments: Clear

Data collection

Comments:

• For interview guide, was it pilot tested or not?

• The duration of the interviews should be put in this section. Please remove the duration of interviews in results section.

• In what interview was data saturation achieved?

• Were transcripts returned to participants for comment and/or correction?

Data analysis

Comments: Clear (Charmaz’ GT -- constructive data analysis).

Study rigor: It is important that study rigor needs to be explained after data analysis (P9) just before your reflexivity.

Data and ﬁndings and reporting

Comments: I suggest that authors indicate each category separately (category 1, category 2, etc.)

References:

Comment: Please refer to the author guidelines for your references

Recommendation:

It is a well written manuscript. However, study methods need to be improved so that the readers can follow the study design. Please see my comments and suggestions.

My recommendation is: “Minor Revision”

I suggest authors to adopt Consolidated criteria for reporting qualitative research (COREQ):

Tong A, Sainsbury P, Craig J. Consolidated criteria for reporting qualitative research (COREQ): a 32-item checklist for interviews and focus groups. International Journal for Quality in Health Care. 2007. Volume 19, Number 6: pp. 349 – 357. Available at: https://academic.oup.com/intqhc/article/19/6/349/1791966

6. PLOS authors have the option to publish the peer review history of their article (what does this mean?). If published, this will include your full peer review and any attached files.

Reviewer #1: No

---

## [Author Response · Author response to Decision Letter 0]

20 Mar 2023

We have outlined our responses to editor and reviewer feedback in the attached response to reviewers letter.

---

## [Editor Report · Decision Letter 1]

28 Mar 2023

Health and social care workers experiences of coping while working in the frontline during the COVID-19 pandemic: one year on.

PONE-D-22-34685R1

Dear Dr. Billings,

We’re pleased to inform you that your manuscript has been judged scientifically suitable for publication and will be formally accepted for publication once it meets all outstanding technical requirements.

Kind regards,

Muhammad Arsyad Subu, Ph.D

Academic Editor

PLOS ONE
---

## [Editor Report · Acceptance letter]

3 Apr 2023

PONE-D-22-34685R1 

Health and social care workers experiences of coping while working in the frontline during the COVID-19 pandemic: one year on. 

Dear Dr. Billings:

I'm pleased to inform you that your manuscript has been deemed suitable for publication in PLOS ONE. Congratulations! Your manuscript is now with our production department. 

Kind regards, 

on behalf of

Dr. Muhammad Arsyad Subu 

Academic Editor

PLOS ONE